# Type of Sex Education in Childhood and Adolescence: Influence on Young People’s Sexual Experimentation, Risk and Satisfaction: The Necessity of Establishing School Nursing as a Pillar

**DOI:** 10.3390/healthcare11121668

**Published:** 2023-06-06

**Authors:** Silvia Navarro-Prado, María Isabel Tovar-Gálvez, María Angustias Sánchez-Ojeda, Trinidad Luque-Vara, Elisabet Fernández-Gómez, Adelina Martín-Salvador, Ana Eugenia Marín-Jiménez

**Affiliations:** 1Department of Nursing, Faculty of Health Sciences, University of Granada, 52071 Melilla, Spain; silnado@ugr.es (S.N.-P.); triluva@ugr.es (T.L.-V.); elisabetfdez@ugr.es (E.F.-G.); 2Department of Nursing, Faculty of Health Sciences, University of Granada, 51001 Ceuta, Spain; matoga@ugr.es; 3Department of Nursing, Faculty of Health Sciences, University of Granada, 18071 Granada, Spain; ademartin@ugr.es; 4Department Quantitative Methods for Economics and Business, Faculty of Economics and Business Administration, University of Granada, 18071 Granada, Spain; anamarin@ugr.es

**Keywords:** adolescent, childhood, internet, personal satisfaction, school nursing, sex education, social networking

## Abstract

The aim is to identify the source of sexuality education used during childhood and adolescence and to analyse whether this education influences their sexual attitudes, their ability to cope with unwanted situations they experience and their satisfaction with their sexual lives. This is a non-experimental, quantitative, ex post facto, cross-sectional study. The sample is formed of 675 young people, with 50% of the ages included being between 20 (Quartile 1) and 22 (Quartile 3) years. The data collection was done by means of an online questionnaire, including sociodemographic and Likert questions about their sex life. Fisher “Independence Contrasts”and correlations were used to see and quantify the relationship among the variables. The main source of education was pornography (29.3%) and the internet (12.4%). The source of education has a significant relationship with whether the use of contraceptives is accepted or not (*p* < 0.001), refusal to use contraceptives (*p* < 0.001), engaging in risky sexual practices (*p* < 0.001), facing unwanted sexual situations (*p* < 0.001) and dissatisfaction with their sex life (*p* < 0.001). It is necessary that children and adolescents have sex education in safe environments, such as in a home or school setting, and the school nurse plays a vital part in this education. This would reduce the need for adolescents and young people to use the internet and pornography as a source of education. School nurses should be the educational axis to offer children and adolescents a reliable point of information about sex education. A joint work with teachers, nurses, students and parents would contribute positively to reduce the number of risky situations young people are facing, and it would promote and improve healthy attitudes towards sex and interpersonal relationships.

## 1. Introduction

Health education has proven to be an effective tool both in disease prevention and promoting health. Through it, health workers provide at-risk groups with specific strategies for self-care [1,2].

The traditional approach used in sexual education campaigns revolved around basic concepts of anatomy, physiology and prevention of risky sexual practices. Furthermore, these sex education campaigns have usually concentrated on avoiding two of the most prevalent problems among young people: prevention of sexually transmitted diseases (STDs) and unintended pregnancies [3,4,5]. It is observed that, although sex education programmes are being offered to adolescents, they do not seem to be effective, as there is no decrease in STDs or unintended pregnancies, and risky sexual practices are increasing [6,7]. According to the WHO [8], approximately 16 million adolescents aged 15–19 give birth each year, and certain STD infections, such as chlamydia, gonorrhoea, syphilis and trichomoniasis, are on the rise. Saturation with respect to traditional sex education campaigns is considered one of the main risk factors for unsafe sexual behaviour [9].

In terms of sexual health, risk groups have traditionally been considered to be those who, due to different circumstances, have been classified as vulnerable [10]. Thus, adolescents, those in the LGBTQ community, and sex workers have been the groups that formed the main focus of sex education campaigns, leaving children and older adults, as well as heterosexual adults in general, partially forgotten [11].

Even though adolescents are one of the groups that sexual campaigns are focused on, some research has shown their lack of knowledge and their low perception of risk. Its consequences are that they have unprotected sexual intercourse with different partners, endangering their health [12].

Furthermore, many adolescents do not view the family as a source of sexual information, as many parents are uncomfortable discussing sexuality with their children. All these aspects lead many adolescents and young people to turn to their peers or to the internet for education or to resolve any doubts they may have about sexuality. The easy access to sexual content offered by social networks and new technologies puts adolescents at risk; their need to seek out new experiences, their impulsivity and sense of impunity only serve to place them in a situation of vulnerability to risky sexual practices [13]. Family and health professionals should be the main affective-sexual pillar since it would avoid the need for young people to look for information on sources that can be a danger to their development.

The combination of poor sex education, lack of family support and easy access to sexual content on the internet creates a context in which adolescents increasingly access pornography, conditioning their sexual learning to the materials viewed [14].

Pornography is undoubtedly on the rise, underpinned by the aforementioned ease of access and the diversity of sexual practices it offers. If used as a source of sex education, it can be detrimental to gender roles, creating a false belief among young people of female subjugation and male aggressiveness, prioritising male pleasure and exaggerated female expressiveness. Pornography is also characterised by risky behaviours, such as condomless sex, group sex and humiliation. If an individual starts viewing pornography at an early stage when they are still developing empathy and affectivity, it causes a distortion in the interpretation of what is real or what is theatrical in sexual relations among young people, where affectivity is mostly absent [15]. It is therefore necessary to equip young people with the knowledge, attitudes and skills to make responsible decisions in their affective-sexual relationships.

Hence, it is essential that programmes are modified and that the content that has traditionally formed part of sex education is updated. In light of digital influence, it is urgently required to include aspects of affectivity, interpersonal relationships and empowerment.

It is therefore important to understand young people’s knowledge and attitudes, as well as the relationship between sexual practices and the sources used to obtain sexuality education in order to reduce the consequences of unsafe sexual practices and gender inequalities.

Therefore, the objective of this study is to ascertain whether the source of sex education used during childhood and adolescence influences young people’s sexual attitudes and to analyse whether this education influences their ability to deal with unwanted situations they experience and their satisfaction with their sex life.

## 2. Materials and Methods

### 2.1. Study Design

This is a non-experimental, quantitative, ex post facto, cross-sectional study. An online, cross-sectional survey was conducted anonymously by self-administration. The variables studied are divided into:−Sociodemographic variables: age (years old), sex (women/men), gender (female/male/non-binary).−Variables related to sex education and attitude towards sexual intercourse: source of sex education; where they would go to resolve doubts; age at which they started viewing pornography; age at first sexual intercourse; relationship with first sexual partner; use of contraceptive method at first and last sexual intercourse and, if not used, reason for not having used it.−Variables relating to sexual practices: karezza, bukkake, stealthing, roulette, bondage, cybersex or sexting, Norwegian, donkey punch, cunnilingus, fellatio and chemsex (Nomenclature).−Variables relating to unwanted situations experienced: sexual harassment and abuse, unintended pregnancy, voluntary termination of pregnancy and sexually transmitted diseases.−Variables relating to satisfaction with sex life: not at all satisfied, somewhat satisfied, satisfied, very satisfied and fairly satisfied.

### 2.2. Participants

This study was carried out in an Andalusian university among students of the bachelor’s degrees in nursing and physiotherapy, with a total of 766 and 363 students, respectively. A non-probabilistic convenience sampling was performed with a confidence level of 95% and maximum estimation error of 5%; a sample size of 287 students was estimated. Finally, the sample consisted of 675 university students who studied between 2020 and 2022. The mean age was 21.37 years (SD = 2.88). The 60.6% were women and the 39.4% were men. They identifies themselves as female 59.3 per cent, as male 36.9 per cent and as non-binary 3.9%.

### 2.3. Instrument

An adaptation of the questionnaire developed and validated by Sanz-Martos et al. [16] was used for this study. It is divided into four sections: level of knowledge about contraceptive methods, attitudes towards contraceptive use, amount sexual activity engaged in and knowledge of family planning centres. A Cronbach’s alpha value of 0.71 was obtained. The adaptation used included the questions referring to the section on attitudes towards contraceptives and sexual activity, to which sexual practices and pornography viewing were added, obtaining a Cronbach’s alpha value of 0.814. These were not taken into account in the original questionnaire, but, due to their current importance, were significant for this study. We also asked about unwanted situations experienced by the participants.

### 2.4. Procedure

This information was shown by the researchers who attended classes first and then online because an informative text was included at the beginning of the questionnaire. The members of the research group invited all students enrolled to participate. After the first dissemination of the questionnaire, a reminder was sent.

The students who ultimately agreed to participate were asked to sign an informed consent form and to complete the questions of the instrument as honestly as possible, and at this point, the anonymous and voluntary nature of the instrument was emphasized. Since sexual issues pertain to an individual’s intimacy, the online method was chosen, as this way, the students could complete it at the time and place of their choice.

The criteria for selecting were they had to be enrolled in the Faculty of Health Sciences and to have access to the digital broadcasting platform; the criteria for exclusion were not to signing the informed consent and errors in the completion of the questionnaire.

The informed consent and questionnaire were released via the digital broadcasting platform, giving all the students access to the information. Neither personal data were requested nor answers were linked to any entity, assuring absolute anonymity.

### 2.5. Statistical Analysis

The statistical analysis was carried out using the IBM SPSS 26 software (SPSS Inc., Chicago, IL, USA). The sample was described by means of frequencies and measures of central tendency (mean and standard deviation) pursuant to the quantitative or qualitative nature of the variables analysed. The scale’s normality of values was explored, and as a result of not being possible to verify their parametricity, for the inferential statistics, a nonparametric analysis was adopted. Like most of the variables are measured in a nominal scale, the nonparametric Kruskal–Wallis has been used to compare different groups of a quantitative variable, and Fisher’s exact test was used to look for relationships between two or more qualitative variables. The significance level assumed was *p* < 0.05.

### 2.6. Ethical Considerations

This study complies with the good clinical practice regulations, as stated in the European Directive 2001/20/CE and Law 14/2007 of 3 July on biomedical research. Treatment of personal data in health research is governed by Organic Law 3/2018 of 5 December on Data Protection and Guarantee of Digital Rights. The protocol obtained a favourable resolution from the Biomedical Research Ethics Committee of the province of Granada. Participants were informed of the objectives of the study and provided their consent to participate by checking a specific box.

## 3. Results

### 3.1. Basic Characteristics of the Sample

When analysing the source of sex education used by young people during their childhood and adolescence, 29.3% (198) were educated through pornography, with the average age they started viewing pornography being 13.49 years (SD 2.45) and an average age of 13 (Q_1_ = 12, Q_2_ = 15). Another large percentage, 22.5% (152), received their education through the internet, 12.4% (84) did so through educational talks, 10.7% (72) received their education at school, 10.2% (69) used social networks, 9% (61) were educated by their parents, 3.6% (24) had no education, as they stated that they were educated through their own sexual experience and finally, 2.2% (15) were educated through their friends.

About contraception and sexuality, the vast majority of young people would turn to the internet: 39.7% (268), followed by 31.1% (210), would consult friends, 21% (142) would approach health personnel, 7.6% (51) would rely on parents, and a small percentage, 0.6% (4), would opt for informative talks.

Young people’s attitudes towards contraceptives were investigated, asking about their use of contraceptives during their first and last sexual intercourse, as well as the reason for not using contraceptives, if this were the case. These results are shown in Table 1.

It is observed the male condom is the most widely contraception method used in the first sexual intercourse, decreasing to the use of barrier contraception methods as the least frequently used. The reasons are not to reduce the sexual pleasure and because it is casual sexual intercourse.

Young people were then asked whether they knew or had performed some of the sexual practices most commonly viewed on sexual websites (Figure 1).

### 3.2. Relationship between Variables

In relation to the source of education and its influence on sex life, it was found that boys who were educated by watching pornography started watching it younger than girls (*p* < 0.001), the median for boys is 13 years (Q3–Q1 = 14–11) compared to 15.50 for girls (Q3–Q1 = 17–13). Similarly, a relationship was found between education and later seeking help when faced with a sexual problem (*p* < 0.001).

The source of sex education is also related to attitudes towards one’s first sexual intercourse. Thus, there was a significant difference between the relationship they had with their first sexual partner, the contraceptive method used and, if not used, the reason why (*p* < 0.001). Thus, it was observed that those who used pornography as a source of education were the ones who had their first relationship with strangers, 70% (49) (*p* < 0.001), and either did not use contraception, 57.3% (63) (*p* < 0.001), or used the withdrawal method, 77.8% (91) (*p* < 0.001). The attitude towards the last relationship maintained continues to be significantly related to the source of sex education used (Table 2).

Table 3 provides information on the relationship between the education received and the performance or knowledge of certain sexual practices.

As can be seen in Table 3, all sexual practices (*p* < 0.001) except karezza (*p* = 0.582) are related to the source of sex education. In all cases, it is the young people whose education was based on viewing pornography who are most likely to engage in these practices and, again in all cases, even if they do not put them into practice, they know what they consist of.

Significant differences were also found in the relationship between the source of sex education and the possibility of experiencing or suffering an unwanted situation, as well as with the sex life satisfaction experienced (Table 4).

As can be seen in Table 4, those people who received their education through pornography have experienced the greatest number of unwanted situations, with 66.7% having experienced unintended pregnancies, 67.4% having experienced voluntary termination of pregnancy and 64.8% having experienced sexually transmitted diseases. Similarly, the people who have used pornography for education are the most dissatisfied (57.1%).

## 4. Discussion

The objective of this study is to ascertain whether the source of sex education used during childhood and adolescence influences young people’s sexual attitudes and to analyse whether this education influences their ability to deal with unwanted situations and their sex life satisfaction they experience.

The outstanding results of this study highlight the influence of pornography on the young people’s life. In this respect, we can observe those whose sexual education was based on pornography have more sexual practices with unknown people and less use of contraceptive methods. Furthermore, these young people affirm they practice more risky sexual practices, face with unwanted pregnancies, face sexual transmission diseases and unwanted situations, such as abuse. As a result, these young people report greater dissatisfaction than other young people who do not use porn.

As the results show, during adolescence, the only source of sex education for most young people is viewing pornography, followed by internet and social media consultations, leaving other traditional forms of sex education, such as educational talks, school education and parental conversations, in the background. These findings coincide with those reported in numerous studies, in which viewing sexually explicit scenes tends to be the main source consulted by adolescents in pursuit of formal education [15,17,18]. Among the young people studied, it has been observed that the age they start watching pornography differs between the sexes, with males being the earliest, with a difference of almost three years compared to females, a fact that is also demonstrated in other studies [15,19,20,21].

It has been observed that traditional forms of sex education involving interaction with educators have been the participants’ least used sources, with the use of impersonal sources, such as pornography, the internet and social networks, growing exponentially. Other studies have also shown that adolescents and young people prefer anonymity, not only for education but also for personal interactions [14,15].

Another fact that has been demonstrated, and which coincides with other studies, is that adolescents and young people prefer to turn to the internet or their friends when they have a problem or doubt about sexuality. Yet again, it can be seen that mentors, such as health personnel and parents are less important to young people for support in their sex education [14,15,22].

Regarding the age of onset of sexual relations, the young people studied had their first sexual intercourse at 16.34 years of age, similar to that of numerous international studies [20,22,23,24,25,26,27,28,29,30,31]. Studies suggest that viewing sexual content is a key factor contributing to early sexual intercourse as well as fostering negative attitudes towards contraceptive methods, particularly barrier methods [20,32,33].

In this regard, it has been observed that viewing pornography as a source of education is related to adolescents and young people attitude towards contraceptive use during both their first relationship and during their last coital relationship before taking the questionnaire. Thus, we can see how young people who were educated through pornography were those who used the withdrawal method to a greater extent or used no method, and when they did use a condom, it was only for the moment before ejaculation with no protection used for foreplay [14,20,32,34]. Those who received educational talks are the best protected, not only against pregnancy, but also against sexually transmitted diseases as they used condoms and hormonal contraception, a fact observed internationally [13,35,36].

Similarly, we see how the source of education is also related to the reason given by young people to justify not having protected themselves in their first and last coital relationship. Thus, having an improvised relationship, not making a bad impression and not diminishing pleasure were the reasons given by young people who were educated with impersonal media, such as pornography, social networks or the internet. These motives are similar to those reported by Sanz-Martos et al. [16].

Numerous studies have demonstrated the role of pornography and the internet as facilitators of risky sexual practices. We, therefore, see that all the sexual practices studied are related to the source of education, except Karezza, a fact that can be explained by the fact that this practice has strong affective connotations, a dimension that is absent in the most widely used sources of education among the participants in this study. Again, it has been observed that young people educated through pornography are more likely to engage in these practices or, even if they do not engage in them, are aware of them. We must bear in mind that these sexual practices have a high degree of violence, humiliation towards women and, in some cases, such as stealthing, they have illegal connotations [37,38]. Moreover, authors such as Álvarez [34] reported that young women who regularly consume pornography report having unwanted sex more frequently than men, which they justify by the need to satisfy the man, a common scenario in pornography.

Finally, it was found that the source of sex education is also related to the level of satisfaction that young people say they have in their sex life. Again, we found that the use of pornography leads to higher dissatisfaction levels as compared with other sources of education, as demonstrated by different authors [39,40,41]. This is concerning, as these adolescents may consider pornography to be a good source of sex education, and this distorts the authentic sexual relationship based on affection and sexual communication.

Although the cross-sectional design of the study does not allow us to establish causal relationships, our findings raise hypotheses that may serve as a useful resource for future prospective and multicentre research. In addition, it could be useful in designing socio-educational strategies to address affective-sexual education among adolescents. School nurses have an important role to play in educational centres as professionals who are directly involved in the care and prevention of risky sexual behaviour and violence against women, especially at an early age.

### Limitations

This study has several limitations. Since the subject studied was intimate and, as the questionnaire was completed online, there was a lack of information because not all of them had access to media at home. Moreover, when asked about the formal education (sex education talks, school, parents) received, it was not possible to know the content, number of sessions or orientation of this training. As for non-formal education (internet, pornography, friends, social networks, self-experimentation), it was not possible to know the exact content that was consulted most. Based on all these factors, expanding this research should be convenient, bearing in mind specific information related to the students´ affective-sexual education.

It is also possible that given the sensitive nature of the subject addressed, when asking for personal opinions, there could have been a social desirability bias that was resolved by guaranteeing the anonymous nature of all participants.

## 5. Conclusions

The results of this study have highlighted the lack of formal sex education during childhood and adolescence, instead showing a preference to obtain information through pornography or the internet rather than going to health professionals or their parents. This can lead to dangers for their affective-sexual health, not only because of the risky practices that are displayed but also because they favour sexual practices based on male domination and female submission. Empowering health personnel, especially school nurses, as a source of education and communication is essential to reduce the need for adolescents and young people to take refuge in the anonymity of the internet to learn about affective-sexual aspects. Furthermore, not only school nurses but also the teachers and parents could contribute to educate the students. Moreover, promoting the implementation of school nurses would enrich the joint work with teachers and nurses for an integral affective-sexual education. Healthy attitudes about sex and sexual intercourses would be promoted.

## Figures and Tables

**Figure 1 healthcare-11-01668-f001:**
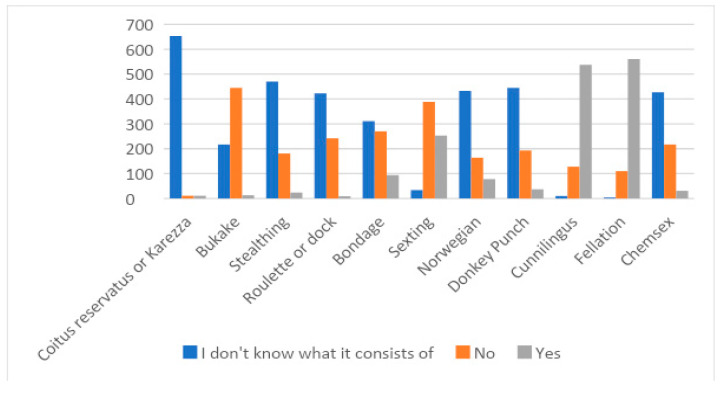
Most visited sexual practices on sexual websites.

**Table 1 healthcare-11-01668-t001:** Contraceptive use and acceptability at first and last sexual intercourse.

	First Sexual Intercourse	*n* (%)	Last Sexual Intercourse	*n* (%)
Contraceptives	Male condom	402 (59.6)	Male condom	289 (4.8)
	Withdrawal method	117 (17.3)	Withdrawal method	79 (11.7)
	Pill	22 (3.3)	Pill	441 (29.9)
	None	110 (16.3)	None	89 (13.2)
	Male condom and hormonal contraceptive	24 (3.6)	Male condom and hormonal contraceptive	35 (5.9)
	Female condom	0 (0)	Female condom	8 (1.2)
	Diaphragm	0 (0)	Diaphragm	5 (0.7)
	Ring	0 (0)	Ring	4 (0.6)
	None in foreplay and condom for ejaculation	0 (0)	None in foreplay and condom for ejaculation	8 (1.2)
Reason for not using contraceptives	To not diminish pleasure	94 (41.3)	To not diminish pleasure	72 (39.3)
	Improvised relationship	78 (34.7)	Improvised relationship	63 (34.4)
	To not give a bad impression	36 (16)	To not give a bad impression	0 (0)
	Trust in the partner	0 (0)	Trust in the partner	24 (13.1)
	Ashamed to buy contraceptives	18 (8)	Ashamed to buy contraceptives	0 (0)

**Table 2 healthcare-11-01668-t002:** Source of sex education related to first and last sexual intercourse and contraceptive acceptance.

		First Coital Intercourse N (%)									Last Coital Intercourse N (%)								
		Internet	School	Sex Education Talks	Parents	Pornography	Friendships	Own Experimentation	Social Networking	*p*	Internet	School	Sex Education Talks	Parents	Pornography	Friendships	Own Experimentation	Social Networking	*p*
Contraceptive method (*n* = 675)	Male condom	110 (27.4%)	61 (15.2%)	60 (14.9%)	49 (12.2%)	29 (7.2%)	12 (3%)	20 (5%)	61 (15.2%)	<0.001	68 (23.5%)	38 (13.1%)	34 (11.8%)	25 (8.7%)	72 (24.9%)	7 (2.4%)	17 (5.9%)	28 (9.7%)	<0.001
	Female condom	0 (0%)	0 (0%)	0 (0%)	0 (0%)	0 (0%)	0 (0%)	0 (0%)	0 (0%)		0 (0%)	1 (12.5%)	0 (0%)	0 (0%)	2 (25%)	0 (0%)	0 (0%)	5 (62.5%)	
	Withdrawal method	8 (6.8%)	4 (3.4%)	2 (1.7%)	8 (6.8%)	91 (77.8%)	3 (2.6%)	0 (0%)	1 (0.9%)		13 (16.5%)	5 (6.3%)	8 (10.1%)	0 (0%)	29 (36.7%)	3 (3.8%)	0 (0%)	21 (26.6%)	
	Pill	2 (9.1%)	3 (13.6%)	6 (27.3%)	2 (9.1%)	8 (36.4%)	0 (0%)	0 (0%)	1 (4.5%)		34 (24.1%)	10 (7.1%)	19 (13.5%)	24 (17.0%)	42 (29.8%)	2 (1.4%)	4 (2.8%)	6 (4.3%)	
	Condom and hormonal contraceptive	4 (16.7%)	1 (4.2%)	9 (37.5%)	2 (8.3%)	7 (29.2%)	0 (0%)	1 (4.2%)	0 (0%)		4 (11.4%)	6 (17.1%)	0 (0%)	2 (5.7%)	12 (34.3%)	3 (8.6%)	0 (0%)	8 (22.9%)	
	Vaginal ring	0 (0%)	0 (0%)	0 (0%)	0 (0%)	0 (0%)	0 (0%)	0 (0%)	0 (0%)		3 (75%)	0 (0%)	0 (0%)	0 (0%)	1 (25%)	0 (0%)	0 (0%)	0 (0%)	
	Diaphragm	0 (0%)	0 (0%)	0 (0%)	0 (0%)	0 (0%)	0 (0%)	0 (0%)	0 (0%)		3 (60%)	0 (0%)	0 (0%)	0 (0%)	2 (40%)	0 (0%)	0 (0%)	0 (0%)	
	None in foreplay and condom in ejaculation	0 (0%)	0 (0%)	0 (0%)	0 (0%)	0 (0%)	0 (0%)	0 (0%)	0 (0%)		3 (60%)	0 (0%)	0 (0%)	0 (0%)	2 (40%)	0 (0%)	0 (0%)	0 (0%)	
	None	28 (25.5%)	3 (2.7%)	7 (6.4%)	0 (0%)	63 (57.3%)	0 (0%)	3 (2.7%)	6 (5.5%)		23 (25.8%)	7 (7.9%)	20 (22.5%)	5 (5.6%)	30 (33.7%)	0 (0%)	3 (3.4%)	1 (1.1%)	
Reason for not using contraception (*n* = 225)	Improvised relationship	16 (20.5%)	0 (0%)	9 (11.5%)	0 (0%)	50 (64.1%)	2 (2.6%)	0 (0%)	1 (1.3%)	<0.001	12 (19%)	0 (0%)	9 (14.3%)	0 (0%)	21 (33.3%)	0 (0%)	0 (0%)	21 (33.3%)	<0.001
	Ashamed to buy contraceptives	6 (33.3%)	5 (27.8%)	0 (0%)	0 (0%)	6 (33.3%)	0 (0%)	0 (0%)	1 (5.6%)		0 (0%)	0 (0%)	0 (0%)	0 (0%)	0 (0%)	0 (0%)	0 (0%)	0 (0%)	
	To not give a bad impression	6 (16.7%)	2 (5.6%)	0 (0%)	1 (2.8%)	23 (63.9%)	0 (0%)	3 (8.3%)	1 (2.8%)		0 (0%)	0 (0%)	0 (0%)	0 (0%)	3 (50%)	0 (0%)	3 (50%)	0 (0%)	
	To not diminish pleasure	6 (6.5%)	0 (0%)	0 (0%)	7 (7.5%)	75 (80.6%)	1 (1.1%)	0 (0%)	4 (4.3%)		16 (22.2%)	7 (9.7%)	10 (13.9%)	8 (11.1%)	27 (37.5%)	3 (4.2%)	0 (0%)	1 (1.4%)	
	Trust in the other person	0 (0%)	0 (0%)	0 (0%)	0 (0%)	0 (0%)	0 (0%)	0 (0%)	0 (0%)		8 (33.3%)	0 (0%)	8 (33.3%)	0 (0%)	8 (33.3%)	0 (0%)	0 (0%)	0 (0%)	

**Table 3 healthcare-11-01668-t003:** Source of sex education related to knowledge or performance of sexual practices.

		Internet	School	Sex Education Talks	Parents	Pornography	Friendships	Own Experimentation	Social Networking	*p*
Karezza	I don’t know what it consists of	148 (22.7%)	71 (10.9%)	81 (12.4%)	58 (8.9%)	190 (29.1%)	14 (2.1%)	24 (3.7%)	67 (10.3%)	0.582
	No	2 (18.2%)	1 (9.1%)	1 (9.1%)	0 (0%)	5 (45.5%)	0 (0%)	0 (0%)	2 (18.2%)	
	Yes	2 (18.2%)	0 (0%)	2 (18.2%)	3 (27.3%)	3 (27.3%)	1 (9.1%)	0 (0%)	0 (0%)	
Bukkake	I don’t know what it consists of	41 (18.9%)	44 (20.3%)	37 (17.1%)	33 (15.2%)	16 (7.4%)	9 (4.1%)	10 (4.6%)	27 (12.4%)	<0.001
	No	110 (24.7%)	28 (6.3%)	47 (10.6%)	28 (6.3%)	117 (39.8%)	6 (1.6%)	14 (3.1%)	35 (7.9%)	
	Yes	1 (7.7%)	0 (0%)	0 (0%)	0 (0%)	5 (38.5%)	0 (0%)	0 (0%)	7 (53.8%)	
Stealthing	I don’t know what it consists of	129 (27.4%)	72 (15.3%)	79 (16.8%)	57 (12.1%)	45 (9.6%)	11 (2.3%)	22 (4.7%)	55 (11.7%)	<0.001
	No	23 (12.7%)	0 (0%)	5 (2.8%)	4 (2.2%)	129 (71.3%)	4 (2.2%)	2 (1.1%)	14 (7.7%)	
	Yes	0 (0%)	0 (0%)	0 (0%)	0 (0%)	24 (100%)	0 (0%)	0 (0%)	0 (0%)	
Roulette	I don’t know what it consists of	117 (27.7%)	65 (15.4%)	80 (18.9%)	57 (13.5%)	25 (5.9%)	12 (2.8%)	22 (5.2%)	45 (10.6%)	<0.001
	No	35 (14.5%)	6 (2.5%)	4 (1.7%)	4 (1.7%)	164 (67.8%)	3 (1.2%)	2 (0.8%)	24 (9.9%)	
	Yes	0 (0%)	0 (0%)	0 (0%)	0 (0%)	9 (100%)	0 (0%)	0 (0%)	0 (0%)	
Bondage	I don’t know what it consists of	82 (26.4%)	44 (14.1%)	47 (15.1%)	41 (13.2%)	29 (9.3%)	12 (3.9%)	20 (6.4%)	36 (11.6%)	<0.001
	No	44 (16.3%)	28 (10.4%)	33 (12.2%)	19 (7.0%)	113 (41.9%)	3 (1.1%)	4 (1.5%)	26 (9.6%)	
	Yes	26 (27.7%)	0 (0%)	4 (4.3%)	1 (1.1%)	56 (59.6%)	0 (0%)	0 (0%)	7 (7.4%)	
Cybersex/Sexting	I don’t know what it consists of	6 (17.6%)	8 (23.5%)	4 (11.8%)	10 (29.4%)	3 (8.8%)	3 (8.8%)	0 (0%)	0 (0%)	<0.001
	No	72 (18.6%)	54 (13.9%)	65 (16.8%)	15 (3.9%)	109 (28.1%)	9 (2.3%)	18 (4.6%)	16 (11.9%)	
	Yes	74 (29.2%)	10 (4%)	15 (5.9%)	36 (14.2%)	86 (34%)	3 (1.2%)	6 (2.4%9	23 (9.1%)	
Norwegian	I don’t know what it consists of	117 (27%)	50 (11.5%)	67 (15.5%)	57 (13.2%)	47 (10.9%)	13 (3%)	20 (4.6%)	62 (14.3%)	<0.001
	No	24 (14.6%)	22 (13.4%)	3 (1.8%)	3 (1.8%)	107 (65.2%)	1 (0.6%)	3 (1.8%)	1 (0.6%)	
	Yes	11 (14.1%)	0 (0%)	14 (17.9%)	1 (1.3%)	44 (56.4%)	1 (1.3%)	1 (1.3%)	6 (7.7%)	
Donkey Punch	I don’t know what it consists of	120 (27%)	52 (11.7%)	63 (14.2%)	55 (12.4%)	59 (13.3%)	12 (2.7%)	21 (4.7%)	63 (14.2%)	<0.001
	No	32 (16.6%)	20 (10.4%)	21 (10.9%)	6 (3.1%)	103 (53.4%)	2 (1%)	3 (1.3%)	6 (3.1%)	
	Yes	0 (0%)	0 (0%)	0 (0%)	0 (0%)	36 (97.3%)	1 (2.7%)	0 (0%)	0 (0%)	
Cunnilingus	I don’t know what it consists of	0 (0%)	1 (10%)	5 (50%)	1 (10%)	0 (0%)	0 (0%)	0 (0%)	3 (30%)	<0.001
	No	29 (22.7%)	21 (16.4%)	14 (10.9%)	23 (18%)	11 (8.6%)	3 (2.3%)	3 (2.3%)	24 (18.8%)	
	Yes	123 (22.9%)	50 (9.3%)	65 (12.1%)	37 (6.9%)	187 (34.8%)	12 (2.2%)	21 (3.9%)	42 (7.8%)	
Fellatio	I don’t know what it consists of	0 (0%)	0 (0%)	0 (0%)	1 (25%)	0 (0%)	0 (0%)	0 (0%)	3 (75%)	<0.001
	No	19 (17.3%)	18 (16.4%)	17 (15.5%)	18 (16.4%)	9 (8.2%)	3 (2.7%)	6 (5.5%)	20 (18.2%)	
	Yes	133 (23.7%)	54 (9.6%)	67 (11.9%)	42 (7.5%)	189 (33.7%)	12 (2.1%)	18 (3.2%)	46 (8.2%)	
Chemsex	I don’t know what it consists of	117 (27.4%)	60 (14.1%)	63 (14.8%)	52 (12.2%)	53 (12.4%)	7 (1.6%)	19 (4.4%)	56 (13.1%)	<0.001
	No	32 (14.7%)	12 (5.5%)	15 (6.9%)	9 (4.1%)	131 (60.4%)	6 (2.8%)	5 (2.3%)	7 (3.2%)	
	Yes	3 (9.7%)	0 (0%)	6 (19.4%)	0 (0%)	14 (45.2%)	2 (6.5%)	0 (0%)	6 (19.4%)	

**Table 4 healthcare-11-01668-t004:** Source of sex education related to unwanted situations and sex life satisfaction.

		Internet	School	Sex Education Talks	Parents	Pornography	Friendships	Own Experimentation	Social Networking	*p*
Risky or unwanted situations	None	102 (21.7%)	69 (14.6%)	50 (10.6%)	57 (12.1%)	99 (21%)	14 (3%)	19 (4%)	61 (13%)	<0.001
	Sexual harassment	18 (29.5%)	0 (0%)	17 (27.9%)	0 (0%)	22 (36.1%)	0 (0%)	2 (3.3%)	2 (3.3%)	
	Sexual abuse	15 (37.5%)	0 (0%)	7 (17.5%)	3 (7.5%)	9 (22.5%)	0 (0%)	3 (7.5%)	3 (7.5%)	
	Unintended pregnancy	1 (33.3%)	0 (0%)	0 (0%)	0 (0%)	2 (66.7%)	0 (0%)	0 (0%)	0 (0%)	
	Voluntary Interruption of Pregnancy	11 (23.9%)	1 (2.2%)	2 (4.3%)	0 (0%)	31 (67.4%)	0 (0%)	0 (0%)	1 (2.2%)	
	STD	5 (9.3%)	2 (3.7%)	8 (14.8%)	1 (1.9%)	35 (64.8%)	1 (1.9%)	0 (0%)	2 (3.7%)	
Sex life satisfaction	Dissatisfied	5 (23.8%)	0 (0%)	1 (4.8%)	2 (9.5%)	12 (57.1%)	0 (0%)	1 (4.8%)	0 (0%)	<0.001
	Somewhat dissatisfied	13 (13.5%)	4 (4.2%)	2 (2.1%)	10 (10.4%)	60 (62.5%)	3 (3.1%)	1 (1%)	3 (3.1%)	
	Satisfied	64 (22.5%)	22 (7.7%)	32 (11.2%)	17 (6%)	109 (38.2%)	6 (2.1%)	10 (3.5%)	25 (8.8%)	
	Very satisfied	48 (25.4%)	33 (17.5%)	30 (15.9%)	24 (12.7%)	13 (6.9%)	3 (1.6%)	8 (4.2%)	30 (15.9%)	
	Fairly satisfied	22 (26.2%)	13 (15.5%)	19 (22.6%)	8 (9.5%)	4 (4.8%)	3 (3.6%)	4 (4.8%)	11 (13.1%)	

## Data Availability

The data presented in this study are available on request from the corresponding author. The data are not publicly available due to privacy.

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
