# Peer review of "Type of Sex Education in Childhood and Adolescence: Influence on Young People’s Sexual Experimentation, Risk and Satisfaction: The Necessity of Establishing School Nursing as a Pillar"

_healthcare, 2023, doi:10.3390/healthcare11121668_

Round 1

Reviewer 1 Report

There are some ambiguities and phrases not well understandable such as:

·       The number of participants (675) is the total number of the students of Nursing and Physiotherapy Departments of the University of Granada which participated in this non-experimental quantitative, ex post facto, cross-sectional study. It would be essential to know the number of students of each Department separately, since it could be giving different results in the particular questionnaire.

·       The gender of the participants isn’t well written to be correctly understandable. What the authors mean with the phrase in page 3, 36.9 (249) considered themselves males and why weren’t identified as males as the females?

·       In the Tables there isn’t mentioned the missing data, since in the limitations of this cross-sectional study, the sample was greatly reduced because was completed online and had an intimate character.

·       The table 3 which provides information about the relationship between the education received and the performance or knowledge of certain sexual practices mentioned in the Appendix A of the article, isn’t reliable because most of the answers refer to the answer  “I don’t know what it consists of”, except the 9th and 10th sexual practice. Also, the number of sample is reduced because is focused on a personal area related to the used sexual practices and also the same comments focusing on the source of sex education related to knowledge or performance of certain sexual practices.

·       The intervention in the normal sexual education of young people of school nurse as an educational axis isn’t analyzed and it could be removed from the title and emphasis should be given in the discussion and conclusions.

Author Response

Dear Reviewer.

We would like to thank you for your suggestions for improving the quality of this article. We have added the proposed modifications, which we hope have responded to your contributions. These are explained in detail below:

  • The number of participants (675) is the total number of the students of Nursing and Physiotherapy Departments of the University of Granada which participated in this non-experimental quantitative, ex post facto, cross-sectional study. It would be essential to know the number of students of each Department separately, since it could be giving different results in the particular questionnaire.
  • Response: It is specified in the Participants section.

  • The gender of the participants isn’t well written to be correctly understandable. What the authors mean with the phrase in page 3, 36.9 (249) considered themselves males and why weren’t identified as males as the females?
  • Response: This has been specified in the text of the manuscript. In this study, participants were asked to self-identify according to sex (female/male) and gender (female/male/non-binary), so we obtained a percentage of females and males who self-identified with the non-binary gender.

  • In the Tables there isn’t mentioned the missing data, since in the limitations of this cross-sectional study, the sample was greatly reduced because was completed online and had an intimate character.
  • Response: A percentage of 0% is added for those variables for which no response was obtained. The target population of 1129, the final sample consists of 675 participants. We believe that the error in the interpretation was the translation of the limitations, which we have already corrected in the text.

  • The table 3 which provides information about the relationship between the education received and the performance or knowledge of certain sexual practices mentioned in the Appendix A of the article, isn’t reliable because most of the answers refer to the answer  “I don’t know what it consists of”, except the 9thand 10th sexual practice. Also, the number of sample is reduced because is focused on a personal area related to the used sexual practices and also the same comments focusing on the source of sex education related to knowledge or performance of certain sexual practices.
  • Response: Table 3 and the annex were included to make it clear to the reader that those whose background was pornography were more aware of and engaged in risky sexual practices than those with more formal training. All sexual practices, except karezza, can put the physical and mental health of the participants at risk. Regarding the sample size, explanation has already been made in previous commentary and modified in text.

  • The intervention in the normal sexual education of young people of school nurse as an educational axis isn’t analyzed and it could be removed from the title and emphasis should be given in the discussion and conclusions.
  • Response: Modification made.

We sincerely hope that we have obtained the necessary quality, we send you our best regards.

For more details please see the revised version manuscript.

Reviewer 2 Report

This study aimed to investigate the influence of sex education sources during childhood and adolescence on young people's sexual attitudes and their ability to deal with unwanted situations, and their sexual life satisfaction. The results revealed that the majority of young people relied on pornography or the internet as primary sources of sex education, which led to risky sexual practices and higher dissatisfaction levels. Traditional forms of sex education, such as educational talks, school education, and parental conversations, were less utilized.

The suggested improvements for the paper can be summarized as follows:

  1. To improve readability and flow, organise the paper more structured, with clear headings and subheadings.
  2. In the Introduction, provide more context on the issue of sex education and its importance, along with a brief overview of the existing literature.
  3. In the Methods section, provide more details on the sampling procedure, data collection, and statistical analysis.
  4. In the Results section, provide an overview of the main findings, use clear and concise language to describe the results, and avoid interpreting or discussing the results at this stage.
  5. In the Discussion section, start by summarizing the main findings, then discuss their implications and compare them with existing literature.
  6. In the Conclusions, provide specific recommendations for improving sex education based on the study's findings.
  7. Throughout the paper, ensure consistent formatting, proper citation of sources, and accurate presentation of tables and figures.

in particular:

abstract:

  1. The abstract could benefit from clearer and more concise language to better convey the study's aim, methodology, and findings. Consider rephrasing some sentences to make them more succinct.
  2. Specify the age range of the young people in the sample, as this will help readers better understand the population being studied.
  3. Provide more information on the methodology, such as the data collection method, types of questions asked, and any statistical tests used in the analysis.
  4. Consider elaborating on the school nurse’s role as an educational axis and how this role may contribute to better outcomes in sex education.
  5. Consider mentioning the practical implications of the study's findings and any recommendations for future research, policy, or practice in the field of sex education.

 Introduction :

  1. Begin the introduction with a broader context, emphasizing the importance of sexual health education and its implications on adolescents' well-being.
  2. Consider reorganizing the paragraphs for better flow and clarity. For example, start by discussing the traditional focus of sex education campaigns, then move on to the problems associated with these campaigns, the role of family support, and the influence of the internet and pornography.
  3. Discuss the specific research gap you aim to address with your study. Explain why understanding the influence of different sources of sexual education is important for improving sex education campaigns and programs.
  4. Provide more information on the potential consequences of inadequate or inappropriate sexual education sources on adolescents' health, relationships, and overall well-being.
  5. Conclude the introduction by clearly stating the objective of your study and briefly mentioning the methodology and sample population.

 Materials and Methods:

  1. Provide a brief rationale for your study design, explaining why a non-experimental, quantitative, ex post facto, cross-sectional study was most suitable for your research objective.
  2. Clearly explain the inclusion and exclusion criteria for participants in your study to ensure that readers can understand how the sample population was selected.
  3. In the description of the instrument, consider providing more information about the original questionnaire's reliability and validity to support its use in your study. Additionally, describe any validation procedures undertaken for the adapted version used in your study.
  4. Provide more details about the procedure of data collection, such as how the online method was facilitated, how participants were contacted, and any measures taken to ensure the confidentiality of the participant’s responses.
  5. In the statistical analysis section, provide more information on the tests used for each variable and any assumptions made or adjustments performed during the analysis.

 Results:

  1. Present the results in a logical order, following the structure of your research questions or hypotheses.
  2. When discussing relationships between variables, provide specific statistical values (e.g., correlation coefficients) to quantify the strength of these relationships and allow for easier comparison.
  3. When discussing the relationship between the source of sex education and unwanted situations or sex life satisfaction, ensure that you provide statistical evidence for these claims (e.g., p-values or chi-square test results) to support your findings.
  4. Consider adding a brief summary of the main findings at the end of the Results section to provide readers with a clear overview of the most important results.

Discussion:

  1. Begin the Discussion section with a brief summary of the main findings from the Results section, highlighting their relevance to the study's objectives.
  2. When comparing your results to findings from previous studies, be sure to provide specific examples, and discuss similarities or differences between your results and those reported in the literature.
  3. Use a more structured approach when discussing the various aspects of your findings, such as the influence of sex education sources, contraceptive use, and sexual practices. This will help guide the reader through your thought process and make it easier for them to understand your arguments.
  4. Clearly discuss the limitations of your study, addressing any methodological issues or potential biases that may have affected the results. Additionally, consider mentioning any confounding factors that could have influenced the observed relationships between variables.
  5. Conclude the Discussion section by highlighting the practical implications of your findings, such as recommendations for improving sex education or addressing the negative impacts of certain sources of sex education (e.g., pornography). The limitations should be stated at the end of the discussion and not after the conclusion. 
  6. Consider discussing possible future research directions that could further explore the relationships between the source of sex education, sexual attitudes, and satisfaction. This may include suggestions for longitudinal or experimental studies or investigations into specific subpopulations or contexts.

Some suggestions for improvement in the Conclusions and Limitations section of the paper:

  1. Begin the Conclusions section with a brief summary of the study’s main findings, emphasizing their relevance to the study's objectives and their implications for sexual attitudes and satisfaction among adolescents and young people.
  2. In the Conclusions section, provide specific recommendations for improving sex education based on your study's findings. For example, discuss the potential benefits of incorporating more comprehensive sex education programs in schools, involving health professionals and parents in the education process, and promoting healthy attitudes towards sex and relationships.
  3. When discussing the limitations of your study (which should be done before the conclusion), it is important to acknowledge the potential impact of these limitations on the interpretation and generalizability of your findings. For instance, consider the extent to which the reduced sample size, online data collection method, and inability to assess the specific content of formal and non-formal education sources may have influenced the observed relationships between variables.
  4. In addition to the limitations already mentioned, consider discussing any other potential issues that may have affected the validity or reliability of your study. This may include factors such as self-reporting biases, sampling biases, or the cross-sectional nature of your study design.
  5. Conclude the paper by summarizing the main points discussed in the Conclusions section, and briefly reiterate the implications of your findings for improving sex education and promoting healthy sexual attitudes and behaviours among adolescents and young people.

The quality of English appears to be good overall. There are some minor errors in grammar, spelling, and punctuation, but these do not significantly detract from the clarity of the writing or the conveyance of the research findings. It is recommended that the authors carefully proofread the manuscript before submission to ensure that any errors are corrected.

Author Response

Dear Reviewer,

First of all we would like to thank you for your suggestions. We hope that the modifications made have increased the quality of the research. We have tried to respond to most of the suggestions you have made. Those that have not been made have been either for reasons of statistical impossibility or because they have not been studied in this research. The modifications are detailed below:

  • To improve readability and flow, organise the paper more structured, with clear headings and subheadings.
  • Response: We have followed the journal's standards for headings and subheadings. We have organised the text for better understanding

  • In the Introduction, provide more context on the issue of sex education and its importance, along with a brief overview of the existing literature.
  • Response: Modification made.

  • In the Methods section, provide more details on the sampling procedure, data collection, and statistical analysis.
  • Response: Modification made.

  • In the Results section, provide an overview of the main findings, use clear and concise language to describe the results, and avoid interpreting or discussing the results at this stage.
  • Response: Modification made.

  • In the Discussion section, start by summarizing the main findings, then discuss their implications and compare them with existing literature. 
  • Response: Modification made.

  • In the Conclusions, provide specific recommendations for improving sex education based on the study's findings.
  • Response: Modification made.

  • Throughout the paper, ensure consistent formatting, proper citation of sources, and accurate presentation of tables and figures.
  • Response: Modification made.

in particular:

abstract:

  • The abstract could benefit from clearer and more concise language to better convey the study's aim, methodology, and findings. Consider rephrasing some sentences to make them more succinct.
  • Response: Modification made.

  • Specify the age range of the young people in the sample, as this will help readers better understand the population being studied.
  • Response: Modification made.

  • Provide more information on the methodology, such as the data collection method, types of questions asked, and any statistical tests used in the analysis.
  • Response: Modification made.

  • Consider elaborating on the school nurse’s role as an educational axis and how this role may contribute to better outcomes in sex education.
  • Response: Modification made. Due to the need to stick to the length, we have made the suggestions summarised.

  • Consider mentioning the practical implications of the study's findings and any recommendations for future research, policy, or practice in the field of sex education.
  • Response: Changes made and adapted also to the suggestion of another reviewer.

 Introduction :

  • Begin the introduction with a broader context, emphasizing the importance of sexual health education and its implications on adolescents' well-being.
  • Response: Modification made.

  • Consider reorganizing the paragraphs for better flow and clarity. For example, start by discussing the traditional focus of sex education campaigns, then move on to the problems associated with these campaigns, the role of family support, and the influence of the internet and pornography.
  • Response: Modification made.

  • Discuss the specific research gap you aim to address with your study. Explain why understanding the influence of different sources of sexual education is important for improving sex education campaigns and programs.
  • Response: Modification made.

  • Provide more information on the potential consequences of inadequate or inappropriate sexual education sources on adolescents' health, relationships, and overall well-being.
  • Response: Modification made.

  • Conclude the introduction by clearly stating the objective of your study and briefly mentioning the methodology and sample population.
  • Response: The introduction concludes with the objective of the research, the methodology and the description of the sample are given in the following section according to the journal's guidelines.

 Materials and Methods:

  • Provide a brief rationale for your study design, explaining why a non-experimental, quantitative, ex post facto, cross-sectional study was most suitable for your research objective.
  • Response: A cross-sectional study was carried out because the objective was to study the extent to which sexuality education has an impact on sexual attitudes.
  •  
  • Clearly explain the inclusion and exclusion criteria for participants in your study to ensure that readers can understand how the sample population was selected.
  • Response: Modification made.

  • In the description of the instrument, consider providing more information about the original questionnaire's reliability and validity to support its use in your study. Additionally, describe any validation procedures undertaken for the adapted version used in your study.
  • Response: Modification made.

  • Provide more details about the procedure of data collection, such as how the online method was facilitated, how participants were contacted, and any measures taken to ensure the confidentiality of the participant’s responses.
  • Response: Modification made.

  • In the statistical analysis section, provide more information on the tests used for each variable and any assumptions made or adjustments performed during the analysis.
  • Response: Modification made.

 Results:

  • Present the results in a logical order, following the structure of your research questions or hypotheses.
  • Response: The structure followed is practically the same as that of the questionnaire, with the only difference being that the results of the first and last sexual intercourse are presented in the same tables, as we thought it would be interesting to show the two moments together.

  • When discussing relationships between variables, provide specific statistical values (e.g., correlation coefficients) to quantify the strength of these relationships and allow for easier comparison.
  • Response: The variables related in the analysis are on a Nominal scale in the majority of cases, so that performing correlation coefficients does not make statistical sense. They have been related by means of Fisher's exact test, since in some cases the conditions for using the Chi-square test were not met.

  • When discussing the relationship between the source of sex education and unwanted situations or sex life satisfaction, ensure that you provide statistical evidence for these claims (e.g., p-values or chi-square test results) to support your findings.
  • Response: Modification made.

  • Consider adding a brief summary of the main findings at the end of the Results section to provide readers with a clear overview of the most important results.
  • Response: This suggestion is also made at the beginning of the Discussion, so it has been considered only to include it in that section in order not to repeat the same information.

Discussion:

  • Begin the Discussion section with a brief summary of the main findings from the Results section, highlighting their relevance to the study's objectives.
  • Response: Modification made.

  • When comparing your results to findings from previous studies, be sure to provide specific examples, and discuss similarities or differences between your results and those reported in the literature.
  • Response: Modification made.

  • Use a more structured approach when discussing the various aspects of your findings, such as the influence of sex education sources, contraceptive use, and sexual practices. This will help guide the reader through your thought process and make it easier for them to understand your arguments.
  •  Response: The discussion is structured in the same order in which the results are presented, in order to facilitate understanding.

  • Clearly discuss the limitations of your study, addressing any methodological issues or potential biases that may have affected the results. Additionally, consider mentioning any confounding factors that could have influenced the observed relationships between variables.
  • Response: The section has been expanded for greater comprehension. We believe that all the limitations found are reflected.

  • Conclude the Discussion section by highlighting the practical implications of your findings, such as recommendations for improving sex education or addressing the negative impacts of certain sources of sex education (e.g., pornography). The limitations should be stated at the end of the discussion and not after the conclusion. 
  • Response: Modification made.

  • Consider discussing possible future research directions that could further explore the relationships between the source of sex education, sexual attitudes, and satisfaction. This may include suggestions for longitudinal or experimental studies or investigations into specific subpopulations or contexts.
  • Response: Suggested changes included in limitations.

Some suggestions for improvement in the Conclusions and Limitations section of the paper:

  • Begin the Conclusions section with a brief summary of the study’s main findings, emphasizing their relevance to the study's objectives and their implications for sexual attitudes and satisfaction among adolescents and young people.
  • Response: Suggestion made in the Discussion section.

  • In the Conclusions section, provide specific recommendations for improving sex education based on your study's findings. For example, discuss the potential benefits of incorporating more comprehensive sex education programs in schools, involving health professionals and parents in the education process, and promoting healthy attitudes towards sex and relationships.
  • Response: This analysis is presented during the article. Teachers and parents are added.

  • When discussing the limitations of your study (which should be done before the conclusion), it is important to acknowledge the potential impact of these limitations on the interpretation and generalizability of your findings. For instance, consider the extent to which the reduced sample size, online data collection method, and inability to assess the specific content of formal and non-formal education sources may have influenced the observed relationships between variables.
  • Response: The sample size was representative.  As we commented in a previous suggestion, we believe that the error was the explanation given initially, as the sample was representative of the population under study. Explanations given above. Suggestion made.

  • In addition to the limitations already mentioned, consider discussing any other potential issues that may have affected the validity or reliability of your study. This may include factors such as self-reporting biases, sampling biases, or the cross-sectional nature of your study design.
  • Response: Made in limitations.

  • Conclude the paper by summarizing the main points discussed in the Conclusions section, and briefly reiterate the implications of your findings for improving sex education and promoting healthy sexual attitudes and behaviours among adolescents and young people.
  • Response: Modification made.

Once again, we would like to reiterate our thanks and hope that the modifications will be as you expected.

Best regards.

For more details please see the revised version manuscript.

Reviewer 3 Report

The claim made in the conclusion about the value of a school nurse as a source of or involved in sexuality education is not necessarily the only conclusion that can be made. It is an equally valid point that teachers can also be trusted professional within sexuality education.  Additionally school nurses are not a common feature in all schools where as teachers are. Rather than continuing to locate sexuality education within a health context there is considerable value in ensuring that teachers are provided with relevant professional development to undertake the described sexuality education.

Author Response

Dear Reviewer.

First of all, we would like to thank you for your input and your suggestion has been included in the text of the manuscript. The role of teachers and parents has been added, and the teamwork of all of them for education has been emphasised.

We hope that the modifications made, not only in the Conclusion section, have achieved the proposed quality.

Thank you again for your contribution.

Yours sincerely.

Round 2

Reviewer 1 Report

No comments, only some minimal mistakes, such as; in page 2, there is a paragraph in Spanish language in red letters in the text  (lines 55-58) and must be removed and also in page 4, there is missing the letter e in word the (line 136) and the word questionnarie isn't correct in lines 136  and 137, and should be questionnaire. Also, in the same page the word que in red in line 137 should be removed.    

The authors have made the appropriate changes and corrections in the revised version of their non experimental quantitative, ex post facto, cross-sectional study.

Author Response

Dear Reviewer,

Thank you very much for your comment. We have made the changes indicated.

Autors.

Reviewer 2 Report

Dear Authors,

I express our sincere gratitude for considering the suggested improvements we provided for your research paper. We appreciate your openness to feedback and willingness to enhance your study’s quality and impact.

In addition to the previous suggestions, I would like to offer further recommendations to strengthen your paper.

Firstly, please provide a more detailed explanation of the methodology employed in your study. Clarifying the sampling procedure, data collection instruments, and data analysis techniques would improve the transparency and replicability of your research. Including information on the reliability and validity of your measures, as well as any steps taken to address potential biases, would further strengthen the methodological rigour of your study.

Secondly, it would be valuable to expand upon the theoretical framework or conceptual model that underpins your research. Presenting a clear theoretical foundation would help readers understand the rationale behind your hypotheses and research questions. Additionally, integrating relevant theoretical perspectives or frameworks from the field of sex education or psychology would enrich the interpretation of your findings.

Furthermore, I suggest incorporating a section on the ethical considerations of your study. Discussing how you addressed ethical concerns, such as informed consent, confidentiality, and participant well-being, would demonstrate your commitment to ethical research practices. If any ethical challenges or dilemmas were encountered during the study, it would be insightful to share your experiences and the strategies employed to mitigate these issues.

Lastly, consider the practical implications of your research findings. How can the results of your study inform the development of sex education programs or interventions? Based on your findings, what recommendations can be made to educators, parents, policymakers, or healthcare professionals? Providing concrete suggestions and actionable recommendations would enhance the real-world impact of your research.

I want to acknowledge the value and significance of your work in shedding light on the influence of sex education sources on young people's attitudes and behaviours. By incorporating these additional improvements, your paper will further contribute to the field and guide future research and practice in sex education.

Thank you for being so dedicated to advancing knowledge in this important area. I eagerly await the revised version of your paper and look forward to witnessing the positive impact it will have on the field of sex education.

With warm regards,

The quality of the English language in the paper is generally good. The authors have demonstrated a solid command of the language and have effectively conveyed their ideas and findings. The writing is clear, concise, and coherent, making it easy to follow the flow of the paper.

I suggest that attention to grammar and sentence structure would help eliminate minor errors or inconsistencies. Proofreading the paper for grammatical accuracy, subject-verb agreement, and proper punctuation would contribute to the overall readability of the manuscript

Author Response

Dear Reviewer,

Thank you very much for taking the time to review our manuscript. Your comments have encouraged us to improve the manuscript and to correct several important omissions from the previous version. We respond to your comments point by point. In the revised manuscript, substantive changes have been highlighted in red.

  • We have expanded the methodology section by providing more information on the study and added a section on ethical considerations. See section Materials and Methods.
  • Information has been added to the theoretical framework to justify the research question.
  • In addition, we have added a section on implications for clinical practice in the discussion section.

We would like to inform both the Reviewers and the Editor that a native speaker has checked the new version of the manuscript.